# Analysis of Pathogens of Urinary Tract Infections Associated with Indwelling Double-J Stents and Their Susceptibility to *Globularia alypum*

**DOI:** 10.3390/pharmaceutics15102496

**Published:** 2023-10-19

**Authors:** Khaireddine Bouassida, Manel Marzouk, Sahar Nouir, Rim Ghammem, Wissal Sahtout, Meriam Ghardallou, Neila Fathallah, Jalel Boukadida, Mehdi Jaidane, Raoudha Slim, Amira Zaïri

**Affiliations:** 1Urology Department, Sahloul Hospital, Sousse 4054, Tunisia; bouassidakhaireddine@gmail.com (K.B.); mjaidane@yahoo.fr (M.J.); 2Laboratory of Microbiology, Hospital Farhat Hached, Faculty of Medicine of Sousse, University of Sousse, Sousse 4002, Tunisia; mnmarzouk@gmail.com (M.M.); jjboukadida@yahoo.fr (J.B.); 3Laboratory of Biochemistry, Faculty of Medicine of Sousse, University of Sousse, Sousse 4002, Tunisia; sanouir@yahoo.fr; 4Department of Epidemiology and Preventive Medicine, University of Sousse, Sousse 4002, Tunisia; ghammam.rim2013@gmail.com; 5Nephrology Department, Sahloul Hospital, Sousse 4054, Tunisia; wassoula78@yahoo.fr; 6Department of Community Medicine, Research Laboratory LR12ES03, Faculty of Medicine of Sousse, University of Sousse, Sousse 4002, Tunisia; meriamelghardallou@yahoo.fr; 7Department of Clinical Pharmacology, Faculty of Medicine of Sousse, University of Sousse, Sousse 4002, Tunisia; neilafathallah@gmail.com (N.F.); raoudha_slim3@yahoo.fr (R.S.)

**Keywords:** ureteral double-J stents, *Globularia alypum*, antibacterial, antibifilm, *Acinetobacter baumanii*, *Enterococcus faecalis*, *Pseudomonas putida*, urinary tract infections

## Abstract

Ureteral double-J stents are frequently used to prevent urinary obstruction. They can develop bacterial colonization and encrustation, which leads to persistent infections that seldom respond to antibiotic treatment. Thus, the goal of this study was to evaluate the local spectrum of bacterial pathogens and their susceptibility to natural compounds. A total of 59 double-J ureteral stents from 59 consecutive patients were examined. The samples were inoculated on agar culture mediums. Extracts of *Globularia alypum* L. were evaluated for their antibacterial activity with the diffusion and broth dilution methods; for antibiofilm activity, the crystal violet assay was used. The identification and the quantification of the different constituents of extracts were determined by reverse-phase high-performance liquid chromatography (RP-HPLC). Bacterial growth was found in three patients (5.1%). *Enterococcus faecalis* (1.7%), *Acinetobacter baumanii* (1.7%), and *Pseudomonas putida* (1.7%) strains were more commonly detected. They were resistant to several common antibiotics. All extracts presented several components, mainly nepetin-7-glucoside and trans-ferulic-acid, and they had antibacterial activity (MIC = 6.25 mg/mL and MBC = 6.25 mg/mL), and antibiofilm (59.70% at 25 mg/mL) properties, especially against *Acinetobacter baumanii*. The results achieved confirm the important role of this plant as a source of therapeutic activities.

## 1. Introduction

The prevention of complications after endoscopic or open urological operations and the management of upper urinary tract obstruction are commonly resolved using ureteral double-J (DJ) stents. They are also utilized to minimize the obstruction risk caused by stone fragments after extracorporeal shock-wave lithotripsy in patients with large kidney stones [1]. The insertion of medical devices into the urinary tract, such as ureteral stents, ureteral double-J (DJ) stents, and urinary catheters, are extensively involved in the management of upper urinary tract obstruction and the prevention of complications after endoscopic procedures or operations open urological pathways [2]. However, their use leads to several health problems, such as dysuria, morbidity, lumbar or suprapubic pain, and hematuria [2,3,4,5]. The insertion of medical devices is linked to more severe issues, such as stent migration, fragmentation, encrustation, and urinary tract infections. The most typical nosocomial infections are urinary tract infections (UTIs) [6]. They cause sepsis, bacteremia, pyelonephritis, renal decline, and even mortality [7]. This has to do with the development of microbial biofilms and the encrustation of urine components that crystallize on the surface of biomaterials.

Biofilms are microbial populations of sessile microorganisms formed by cells that are immersed in a matrix of extracellular polymeric substances linked to an interface or substratum. Bacteria forming biofilm differ physiologically and phenotypically from planktonic or suspended cells [8,9]. During the formation of biofilm, sessile cells acquire physiological characteristics that are different from those of planktonic cells. They will be arranged in macrocolonies surrounded by a matrix of extracellular polymeric substances (EPS) and separated by water channels ensuring the diffusion of oxygen, nutrients, and waste products. EPS, also called matrix, are made of carbohydrates, proteoglycans, DNA, and other biological components, which not only prevent diffusion due to steric hindrance but also may carry charges that could prevent the diffusion of other antibiotics. Bacteria within biofilms will have different growth rates and gene expression [8,9,10]. Biofilm formation on stent colonization was well defined; a few hours are sufficient for the bacterial colonization of DJ stents as they become covered by host proteins, which facilitate bacterial adhesion [8]. The first step is characterized by the formation of a conditioning film with extracellular molecules. Inflammatory peptides, fibrinogen, and blood proteins seem to be involved in biofilm formation. Secondly, other proteins, such as fibrinogen, albumin, and collagen, allow the attachment of microorganisms. Finally, biofilm formation is allowed by bacterial growth under conditional film [9,10]. Bacterial biofilms have been associated with several infections, particularly in patients with indwelling medical devices, such as prostheses and catheters. Recently, they have been described as important mediators of health-care-associated infections and associated with 80% of all chronic human infections [10,11]. Additionally, urinary tract infections frequently develop resistance to biocidal, antibacterial, and immunological stresses, which makes it difficult and occasionally impossible to treat them with traditional chemotherapeutic treatments [12]. These issues have led to a continuing search for alternative treatments for biofilms and planktonic bacteria. Of particular interest, natural compounds, especially plant extracts, offer several properties that might present promising opportunities for the development of new antibiofilm agents [13]. In fact, traditional remedies have been shown to be successful in the treatment of a wide spectrum of chronic diseases since ancient times [14]. There is a significant need for bioprospecting on natural sources of antioxidant chemicals to address these chronic illnesses [15]. *G. alypum* is a wild plant found across the Mediterranean that belongs to the *Globulariaceae* family [16]. Found in Tunisia and named Zrigua, it is one of the most used traditional plant remedies. Its leaves have long been used as hypoglycemic, laxative, and purgative agents. It also contains antioxidants, antiobesity, antihyperglycemic, antihyperlipidemic, antipyretic, and analgesic properties [17]. Considering that the presence of antibiotic-resistant forms of bacteria after urinary tract stenting can lead to the development of multiple complications in urological patients, we aimed, firstly, to investigate the incidence of bacterial colonization on DJ stents and the responsible microorganisms in patients with a negative urine culture (UC) to determine the clinical role of stent colonization in clinical practice. Then, we evaluated the susceptibility of the local spectrum of bacterial pathogens to *G. alypum* extracts to optimize antimicrobial therapy after upper urinary tract stenting.

## 2. Materials and Methods

### 2.1. Ethical Approval

This study complied with ethical standards for research. The Declaration of Helsinki of the World Medical Association on the Ethical Principles of Medical Research involving Human Subjects and Tunisian legal regulations in the field of clinical trial regulation were both consulted regarding the conduct of all research procedures. All study participants signed written informed consent forms after receiving information about this study’s objectives. Patients were only included in the trial willingly and were given the option of participating or not. The database was filled with information on every participant and the identifiers were encrypted. All data gathered throughout this study will be kept totally confidential and used only for research. Ethical approval for this study was obtained from the Bioethics Committee of the Faculty of Medicine of Sousse.

### 2.2. Urine and Double-J Tube Sample Acquisition and Study Design

This prospective study was carried out following approval from the institutional ethical committee. Between January 2023 and May 2023, DJ stents removed from 59 patients were evaluated for the presence of bacteria. Patients with conventional, polyurethane DJ stents that were inserted unilaterally or bilaterally and had negative UC results were included. The placement indication for the DJ stent was noted. Patients with immunological suppression, chronic renal insufficiency, or diabetes mellitus were excluded from the trial. Patients who also had kidney stones were also eliminated, as were those who had previously undergone antibiotic therapy for a UTI. Before the stent removal procedure, none of the patients received prophylactic antibiotics. Each patient provided midstream urine samples several days prior to the removal of the stent. The DJ stents were removed using grasping forceps that were either 15.5 or 22 French rigid under local anesthetic and aseptic conditions. The removed DJ stents were rinsed in a sterile saline solution once divided into three parts (upper, middle, and outer surface) and stored in 50 mL sterile centrifuge tubes containing 2 mL of physiological saline solution at 4 °C for a maximum of 5 days until analysis and microbiological investigation.

### 2.3. Bacterial Culture Identification

All segments of the stent parts, including the inner surface, middle, and outer surface, were cultured according to the technique described by Brun-Buisson et al. [18], adapted from Cleri et al. [19]. In brief, 1 mL of sterile water (physiological serum) was added to the tube containing the corresponding segment. The contents were mixed and vortexed for 1 min to enable the detection of microorganisms attached to the outer surface of the stent segment. Then, 10 µL was sampled and plated on the surface of fresh blood agar and cooked blood agar supplemented with isovitalex. The agar plates were incubated at 37 °C for 48 h for fresh agar and at 37 °C with 5% CO_2_ for cooked blood agar. If colonies were observed, the number was recorded. The culture was considered positive when bacterial growth was observed. An identification and antimicrobial susceptibility assessment were conducted according to standard laboratory methods. Isolated bacteria were initially identified by conventional biochemical methods: Api 20E, Api Staph, and Api Coryne V.2 (BioMerieux1, Marcy-l’Etoile, France).

### 2.4. Antimicrobial Susceptibility Assays

For Gram-negative strains, the minimum inhibitory concentrations (MICs) for penicillins (ampicillin, ticarcillin), β-lactam combination agent (Piperacillin-Tazobactam), cephems (cefotaxim, ceftazidime, caphalothin), carbapenems (ertapenem, imipenem), aminoglycosides (gentamicin, tobramycin, amikacin), quinolones and fluoroquinolones (nalidixic acid, ciprofloxacin, levofloxacin, ofloxacin), nitrofurans (nitrofurantoin), and folate pathway antagonists (trimethoprim/sulfamethoxazole) were determined by the Vitek 2 System according to the Clinical and Laboratory Standards Institute (CLSI) guidelines [20]. For Gram-positive strains, the MICs for 12 drugs, including tetracycline (doxycycline), penicillin (penicillin), cephem (cefotaxime), aminoglycosides (kanamycin), macrolide (erythromycin), lincosamide (clindamycin), oxazolidinone (linezolid), fluoroquinolones (ciprofloxacin and moxifloxacin), glycopeptide (vancomycin), lipopetide (daptomycin), and ansamycin (rifampicin), were determined by micro-dilution in cation-adjusted Muller Hinton broth in accordance with CLSI guidelines [20]. A multidrug resistance (MDR) strain was defined as showing acquired non-susceptibility to at least one agent in three or more antimicrobial categories [21].

### 2.5. Plant Material

The plant material used in this study conform to appropriate national, institutional, and international regulations and rules. In January 2019, *G. alypum* was collected during the full blooming stage in Tunisia’s Ouardanin area (Longitude: 10°40′35″ E; latitude: 10°40′35″ N and altitude: 75 m). The taxonomic identification was approved by Prof. Fethia Harzallah Skhiri of Tunisia’s High Institute of Biotechnology in Monastir. The plant voucher specimen was cataloged as Ga 022 in the Herbarium of the Laboratory of Bioresources: Integrative Biology and Valorization (ISBM). The leaves were washed and dried at room temperature for 7 days until they reached a constant weight.

### 2.6. Preparation of the Infusion

A shade-dried plant (5 g) was soaked for 30 min in 100 mL of boiling distilled water. The infusion was then filtered through Whatman No. 4 paper, lyophilized through sublimation, and stored as a solid powder in an amber glass container at 4 °C until use.

### 2.7. Chemical Analyses of the General Composition of G. alypum Aqueous Extract Using Reverse-Phase High-Performance Liquid Chromatography (RP-HPLC)

The chromatographic analyses of *G. alypum* aqueous extract were carried out using a RP-HPLC device of the Agilent 1200 type controlled by a computer. For the separation, the Kinetex Evo C18 reversed-phase analytical column was used at room temperature. Mobile phase A was formed by 99% water and 1% of formic acid and mobile phase B was formed by a combination of acetonitrile and formic acid (1%). Under these conditions, the pressure developed by the pump was around 150 bars. The flow rate was 1 mL/min. The gradient program was displayed in the following way: 90% A, 10% B (0 min), 80% A, 20% B (20 min), 75% A, 25% B (30 min), 65% A, 35% B (40 min). The sample was filtered through a 0.20 µm filter before injection. The injection volume was 20 µL and peaks were monitored at 324 nm. Peak identification was conducted by comparing the retention time and the UV spectra of *G. alypum* phenolics chromatogram with those of available standards. Quantification was performed by reporting the measured integration area in the calibration equation of the corresponding standard.

### 2.8. Antibacterial Activity

To evaluate the antibacterial effect of *G. alypum* extract against the following strains, *Enterococcus faecalis*, *Pseudomonas putida*, and *Acinetobacter baumanii*, the agar disc diffusion and dilution methods were performed [22]. With a sterile cotton swap, the pathogenic bacterial inoculums were streaked onto Muller-Hinton (MH) agar plates after being adjusted to 0.5 McFarland standard turbidity. The agar medium was covered with sterile filter discs (diameter 6 mm, Biolife, Milan, Italy) and 20 µL of our sample was deposited onto each disc. The lyophilized extract was diluted in sterile distilled water at a concentration of 25 mg/mL. As a standard antibiotic, imipenem (30 µg/mL; 10 µL/disc) was employed. The antibacterial activity was assessed by measuring the inhibition zone that formed around the disc following a 24-h incubation period at 37 °C. The results of each experiment were run in triplicate. Serial dilutions of the extract (0.05–25 mg/mL) were applied to plates with 96 U-bottomed wells (Nunc, Roskilde, Denmark) together with MH broth and the target bacteria; after 24 h of incubation at 37 °C, the turbidity was evaluated by the naked eye. The MIC is the lowest concentration of the extract which prevents the appearance of the visible growth of a microorganism. The MBC was assessed by transferring 10 µL from the well that showed no bacterial growth following the MIC assay on MH agar after incubation at 37 °C for 24 h. The bacterial growth was analyzed after 24 h of incubation at 37 °C and the MBC was reported to be the lowest concentration of the sample that was bactericidal. Each assay was carried out three times. Imipenem (30 µg/mL; 10 µL/disc) was used as a positive control while wells containing distilled water served as negative controls.

### 2.9. Antibiofilm Assay

The crystal violet assay was used to determine the antibiofilm assay [23]. Different dilutions and concentrations of the extracts were prepared in sterile distilled water (0.05–25 mg/mL). Pathogenic bacteria suspension (grown in brain heart infusion (BHI) for 24 h at 37 °C; 10^5^ CFU/mL) were then mixed with the prepared extracts on plates with 96 U-bottomed wells (Nunc, Roskilde, Denmark) containing BHI (Oxoid) with 2% glucose (*w/v*). BHI with 2% glucose was used as negative control while wells containing BHI with 2% glucose inoculated with the pathogenic strain served as positive controls. The plates were then rinsed 3 times with PBS after incubation at 37 °C for 24 h. Cells in the biofilm were fixed with methanol for 15 min, air-dried, and stained with 1% crystal violet. The quantification of biofilm formation was evaluated by measuring the absorbance at 595 nm using a microplate reader (GIO. DE VITA E C, Roma, Italy). Three independent experiments were performed. The percentage of inhibition was calculated by following the following formula: Inhibition (%) = (ODcontrol − ODExtract)/ODcontrol × 100.

### 2.10. Statistical Analysis

Data capture and analysis were performed using the IBM Statistical Package for the Social Sciences version 25. Descriptive statistics were reported as frequencies for categorical variables and as means and standard deviations (SDs) or medians and inter-quartile ranges (IQRs) for quantitative ones. Differences between groups were examined using the Chi-squared (χ^2^)-test to compare proportions. When this test was not applicable, the Fisher test was used.

## 3. Results

### 3.1. General Characteristics of This Study

A total of 59 patients [31 (52.5%) men; 28 (47.5%) women] were included in this study. The mean patient age was 58.41 (±15.54) years. Patient characteristics are presented in Table 1. Overall, 13 (4.2%) had dependent functional capacity and 23 (7.5%) had indwelling urethral catheters due to persistent bladder distention. The mean duration of the placement of ureteral stents was 133.2 (±107.4) days. The primary reasons for DJ stent placement were pyelonephritis in forty-seven (79.7%), retroperitoneal fibrosis in five (8.5%), tumor pathology in five (8.5%), and ureteropelvic junction in two (3.4%) patients.

### 3.2. Bacterial Pathogens in DJ Stents

Regarding the results of urine culture, bacteriuria was found in three patients (5.1%). Two Gram-negative bacilli and Gram-positive cocci bacteria were identified. *Enterococcus faecalis* was isolated (one patient), followed by *A. baumanii* (one patient); *Pseudomonas putida* was detected in the last patient. Bacterial colonization was detected in all parts (upper, middle, and lower) (Table 2). The isolated bacteria showed resistance to most tested antibiotics. For instance, *E. faecalis* isolates were resistant to chloramphenicol, validixine-quinolone, Gentamycine, colistine, cefoxitine_C2G- Cephamycine, Cefazoline_C3G, and cefoxitine_C2G-Cephamycine. *Acinetobacter* isolates were especially resistant to colistine. *P. putida* was mainly resistant to amoxicillin-clavulanic acid. It is important that the identified bacteria were resistant to the most widely used and available antibiotics in Tunisia, which leads to a major public health issue. Several studies reported that during the last decades, *A. baumannii* has emerged as a major pathogen in vulnerable and critically ill patients. Bacteremia, pneumonia, urinary tract, and skin and soft tissue infections are the most common presentations of *A. baumannii*, with attributable mortality rates approaching 35% [24]. *E. faecalis* can cause a variety of infections, including cystitis, pyelonephritis and catheter-associated UTI, endocarditis, and mixed-organism infections of the abdomen and pelvis. Ampicillin and amoxicillin are the agents of choice for susceptible strains [25]. *P. putida* has been demonstrated to cause a lethal case of bacteremia due to skin and soft tissue infections, which have malnutrition, immobility, and peripheral vascular disease as risk factors [26].

Colonized ureteral DJ stents can be a reservoir for microorganisms giving rise to bacteriuria during stent manipulation [25]. Recently, it has been shown that minutes after the insertion of the stent, depositions of the host urinary components formed a conditioning film on the stent [26,27]. It is important to mention that urinary pH plays an important role. Urease, an enzyme produced by bacteria, such as *Pseudomonas* species, cleaves urea to ammonia, which increases urinary pH and results in the precipitation of calcium phosphate crystals and magnesium ammonium phosphate (struvite). These crystals cause the mineralization and incrustation of the biofilm layer on the stent [28]. Bacterial fimbriae and polymeric substances produced by the bacteria are key virulence factors. Microorganisms appear to be more resistant to antimicrobial agents’ biofilms [29]. The bacterial colonization of DJ stents is a main factor in the pathogenesis of stent-associated UTIs; however, the relationship between DJ stents and the development of UTIs remains unclear [30]. Stent colonization does not always lead to bacteriuria. Therefore, negative UC does not rule out a colonized stent [31]. A study elaborated on by Kehinde et al. [32] showed that significant bacteriuria was developed in about 17% of patients with indwelled DJ stents while 42% of patients had their stents colonized. They also mentioned that the UC was sterile in about 60% of their patients with colonized DJ stents. Likewise, Lifschitz et al. [33] found bacteriuria in 15% of their patients and found 45% of their patients to have colonized DJ stents. Our stent colonization rate was 5.1%. However, it is important to note that our study included only patients with negative UC. In many studies, *Escherichia coli* and *Enterococci* spp. were found to be predominant [26,27,28,29]. In our study, Gram-negative bacteria were also found to be predominant in stent colonization (SC). Two recent publications described Gram-positive pathogens, particularly *Staphylococcus* sp., described as the predominant bacterium [26,27,28]. Farsi et al. [34] indicated that *P. aeruginosa* was most frequently isolated from both urine and stents. This difference in isolated microorganisms should be attributed to the variability of countries and institutions. It has been demonstrated that the bacteria isolated from urine before stent placement were less antibiotic-resistant than the germs acquired from indwelling DJ stents. The expression of genes specific to biofilms provided an explanation for this circumstance [29]. Microorganisms may be released into the urine during stent operations or ureteroscopic procedures, which, if SC is positive, could result in sepsis. Therefore, even if the preoperative UC is negative, endoscopic operations following stent removal increase the risk of infectious complications [31]. Immunosuppression, chronic renal insufficiency, and diabetes mellitus have also been shown to be risk factors associated with bacteriuria and stent colonization [29,35]. Additionally, a study conducted by Kehinde et al. [32] showed that these comorbidities were associated with significantly higher rates of stent colonization and bacteriuria. In several studies, age and sex were reported to be significant predisposing factors for stent colonization [36]; however, according to Lifshitz et al. [33], these variables had no impact on the prevalence of bacteriuria and stent colonization. The anatomic proximity of the orifices of the female genital system may provide an explanation for the gender difference. Due to a limited number of patients, we did not find any relationship of SC positively correlating with gender and age.

### 3.3. RP-HPLC Analysis

For the phytochemical characterization of *G. alypum*, a RP-HPLC analysis system was used. In fact, RP-HPLC has several advantages over normal phase HPLC; it has lower costs when compared with other HPLC methods and the used solvents are lower in toxicity. Additionally, RP-HPLC can analyze samples containing polar (hydrophilic), non-polar (hydrophobic), ionic, and ionizable compounds due to the use of a hydrophobic stationary phase [37]. Figure 1 and Figure 2 report the phenolic content of the infusion extract, which turned out to be the most complex one. Based on their retention times, a total of five main compounds were detected and identified. These compounds can be divided into three different phenolic compound classes present in *G. alypum*, namely, phenolic acids (cafeic acid); phenylpropanoid glycoside or nonglycoside (verbascoside and trans-ferulic acid, respectively), and flavonoids glycosides (Nepetin7 glucoside and isorhamnetin 3-o rutinoside). The most abundant compound was represented by nepetin-7-O-glucoside (30.82 µg/mL), followed by trans ferulic acid (10.33 µg/mL) (*w*/*w*) (DW). Caffeic acid and verbascoside were present in smaller quantities (7 µg/mL). The detected compounds have never been reported before in Tunisian *G. alypum* extracts.

The content in the examined extracts was different from the results of other researchers working on Tunisian aqueous *G. alypum* species. In fact, Hajji et al. [38] have shown that by using the HPLC-PDA/ESI-MS analysis, the phenolic fraction of *G. alypum* aqueous extract (GAAE) from northwest of Tunisia was dominated by iridoids and secoiridoids with an abundance of serratoside. They showed that three flavonoids were also detected. Gallocatechin and quercetin glucoside were the main components, followed by phellamurin. The dominant constituent of GAAE was verbascoside. In addition, the work of Bouriche et al. can be cited since they have different compounds in their aqueous extracts. In fact, the authors mentioned that the HPLC-TOF/MS analysis revealed the presence of phenolic acids and flavonoids in *G. alypum* aqueous extracts collected in June 2010 from Sétif in Eastern Algeria. These extracts contain a high quantity of glycoside flavonoids, such as diosmin, rutin, and scutellarin, with a predominant amount of naringin and quercetin-3-β-D-glucoside. Moreover, they revealed that among the determined phenolic acids, the concentration of cinnamic acid was high in the presence of protocatechuic acid [39]. The differences observed in the phenolic profiles of aqueous extracts from various origins could be explained by environmental and growing conditions (soil composition, altitude, rainfall, climate) and the different techniques used to determine this composition. These factors may directly interfere with the content of chemical components. It is likely that our obtained extracts, since they are rich in phenolic compounds, may exhibit a variety of biological activities [40]. Furthermore, as far as we are aware, few studies have been conducted to determine the phenolic composition of aqueous extracts of *G. alypum* in Tunisia. Only a few plants have been evaluated for their aqueous extracts and no information has been previously provided to ascertain the polyphenol composition of decoction or infusion extracts. The majority of investigations that are currently accessible in the literature focused on essential oils or methanolic extracts.

### 3.4. Evaluation of Antibacterial Activity

The ability of aqueous *G. alypum* extracts to inhibit the proliferation of Gram-negative and Gram-positive planktonic cells was assessed and is presented in Table 3 and Table 4. All of the presented bacteria showed sensitivity to the extract, with the most pronounced effect being observed with *A. baumanii* with a zone of inhibition equal to 22.5 mm and a MIC and MBC equal to 6.25 mg/mL. Regarding the diffusion method, the zones of inhibition of vancomycin and infusion appear to be similar. However, zones can always be made to appear similar by adjusting the concentration and volume spotted. No data have been reported before to describe the antibacterial activity of aqueous extracts of *G. alypum*, which makes the discussion limited. However, the few available data concerning this activity reported mostly on the antimicrobial activity of organic extracts [16,41,42,43,44,45]. As an example, we reported previously that extracts of *G. alypum* obtained via sonication showed a significant antibacterial effect on *S. aureus*, with a zone of inhibition of 14.5 mm, which complies with our actual study; however, these methanolic extracts had no effect against the other tested strains [17]. 

According to Nayak et al. [46], the antibacterial effect of the aqueous extracts might be attributed to the anionic components, such as chlorides, thiocyanate, sulfates, and nitrate, along with other water-soluble components that were naturally present in the plant material.

The presence of polyphenols, tannins, and flavonoids in plant extracts might be responsible for the antibacterial activity since these components are generally produced by plants and are involved in their property of defense against microbial infections. Interestingly, our extract is very rich in nepetin-7-glucoside, which is a member of flavonoid and glycoside [45]. In addition, trans-ferulic acid (TFA), presented in abundance in the aqueous extract, has many pharmacological properties, including antimicrobial and antifungal activities [46].

### 3.5. Antibiofilm Activity

Biofilm is a complex matrix of microorganisms in which cells bind together with the extracellular polysaccharides (EPS), which can insulate cells from antibacterial substances and reduce the antibacterial activity [8]. Biofilm-producing bacteria are inherently resistant to antibacterial drugs, which are the major cause of various infections in humans and animals [47]. In the current study, the antibiofilm effect of the infusion extracts of *G. alypum* was tested at different concentrations (MIC and 1/8 MIC). The biofilm inhibition rates for *G. alypum* against *A. baumanii* were from 45.39% to 59.70%; they were from 31.69% to 45.99% when acting against *E. faecalis* (Table 5). Moreover, the biofilm inhibition was weaker for *P. putida*. It is worth to noting that *G. alypum* strongly affected *A. baumanii* and *E. faecalis* biofilm. To the best of our knowledge, it is the first time that the antibiofilm activity of *G. alypum* was reported and no data concerning this activity were previously provided. Additionally, our study revealed that the concentrations equal to or under the 1/4 MIC level of *G. alypum* extract had obvious antibiofilm activity toward *A. baumanii* biofilms. However, our results are in compliance with another study that showed that aqueous extract extracts of miswak were effective against 10 MDR pathogenic organisms, including *A. baumanii*. The various phytochemicals found in medicinal plants, including flavonoids, glycosides, tannic acid, phenolics, and chlorogenic acid, can inhibit or eliminate biofilms by preventing the growth of bacteria that produce them, dissolving the polysaccharides of EPS, rupturing the integrity of the bacteria’s membrane, suppressing the activity of related enzymes, disrupting the fimbriae that bacteria use to adhere to surfaces, and suppressing the expression of biofilm-related genes [48,49]. We speculate that the synergistic effect of phytoconstituents in *G. alypum* might be responsible for antibacterial and biofilm formation inhibition, especially the rich flavonoids and ferulic acid, which have been shown to have antibacterial and biofilm activities. However, the exact mechanisms involved still need to be fully elucidated. We previously showed that our extracts are rich in ferulic acid. Ferulic acid (AF), 3-(4-hydroxy-3-methoxyphenyl)-2-propenoic acid, a secondary metabolite, belongs to the phenolic compound class and is considered one of the most common phenolic acids found in natural species [17].

## 4. Conclusions 

Our study demonstrated that *A. baumanii*, *E. faecalis*, and *P. putidis* were the detected pathogens after stenting 59 patients who were initially abacterial. In addition, the increase in the number of strains after stenting could be due to insufficient adherence to the principles of anti-infective protection. Lack of antibiotic prophylaxis may also contribute to the appearance of bacteria in postoperative patients. The formation of microbial biofilm on the stent surface is due to the indwelling ureteral stents. Bacteria in a biofilm can avoid antimicrobial activity via several mechanisms and phenotypic changes. Further investigations of these mechanisms will aid in identifying therapeutic targets in the treatment of urinary tract infections and urosepsis. The present study also provides a greater insight into the phytochemical composition of *G*. *alypum*. The antimicrobial properties of the investigated extracts provide considerable benefits for health, such as the treatment of diseases related to bacterial infections.

## Figures and Tables

**Figure 1 pharmaceutics-15-02496-f001:**
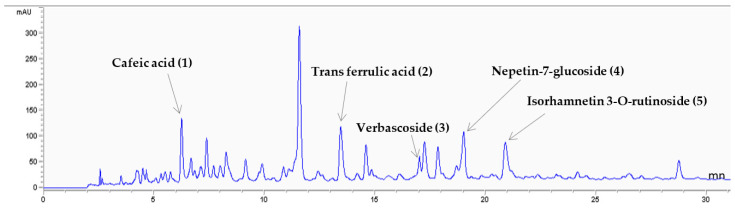
Chromatographic profile of *Globularia*
*alypum* L. extract acquired at 324 nm.

**Figure 2 pharmaceutics-15-02496-f002:**
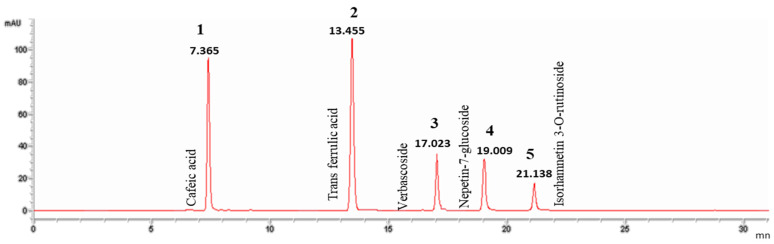
Retention time and chromatographic profile of standards recorded at 324 nm.

**Table 1 pharmaceutics-15-02496-t001:** Patient’s characteristics.

	*n* (%)/SD
Number of patients	59 (100)
Male	31 (52.5)
Female	28 (47.5)
Mean age (years)	58.41 ± 15.54
Colonized stent rate	3 (5.1)
Multiresistant bacteria strains	3 (5.1)
Total number of colonies	3
Mean duration of stenting (days)	133.2 ± 107.4

**Table 2 pharmaceutics-15-02496-t002:** Bacterial strains identified from patients and their antibiotic resistance profile.

Microorganism	N (%)	Antibiotic Profile Resistance
Sterile	56 (94.9)	
*Enterococcus faecalis*	1 (1.7)	chloramphenicol, validixine-quinolone, Gentamycine, colistine, cefoxitine_C2G- Cephamycine, Cefazoline_C3G and to cefoxitine_C2G-Cephamycine
*Acinetobacter baumanii*	1 (1.7)	colistine
*Pseudomonas putida*	1 (1.7)	amoxicilline- clavulanic acid
Fungus	0	

**Table 3 pharmaceutics-15-02496-t003:** The antibacterial activities of aqueous of *G. alypum* against three clinical organisms.

Zone of Inhibition (mm)		
Strains	Imipenem	Infusion
*Enterococcus faecalis*	18.10 ± 0.00	21.00 ± 0.00
*Pseudomonas putida*	19.20 ± 0.53	21.50 ± 1.01
*Acinetobacter baumanii*	17.01 ± 1.02	24.50 ± 0.31

**Table 4 pharmaceutics-15-02496-t004:** MIC and MBC determination of the infusion extract of *G. alypum* leaves. Minimum Inhibitory Concentration (MIC) and Minimum *Bactericidal* Concentration (MBC). N: not found.

Strains	*E. faecalis*	*P. putida*	*A. baumanii*
MIC (mg/mL)	>12	>12	6.25
MBC (mg/mL)	N	N	6.25

**Table 5 pharmaceutics-15-02496-t005:** Percentage of biofilm inhibition after 24 h. Data are the means ± SD of three independent experiments. The values with different superscript letters show significantly (*p* < 0.05) different means.

	Percentage of Inhibition b (%)		
Concentrations (mg/mL)	*P. putida* ^a^	*E. faecalis* ^b^	*A. baumanii* ^c^
25	26.61	45.99	59.70
12.5	18.74	38.38	54.85
6.25	16.49	37.19	53.03
3.12	14.43	31.68	45.39

## Data Availability

The data presented in this study are available within this article.

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
