# Peer review of "Analysis of Pathogens of Urinary Tract Infections Associated with Indwelling Double-J Stents and Their Susceptibility to Globularia alypum"

_pharmaceutics, 2023, doi:10.3390/pharmaceutics15102496_

Round 1
Reviewer 1 Report
Analysis of Characteristics and Pathogens of Urinary Tract Infection Associated with Indwelling Double-J Stent and Their Susceptibility to Natural Compounds.
Bouassida et al
The authors demonstrate the antibacterial and antibiofilm activities in Globularia alypum against pathogens of urinary tract infections. One important aspect of this paper is that aqueous extract from the plant has been used while majority of previous investigations have focused on essential oils or methanolic extracts. The paper makes a significant contribution to the field. I have the following additional comments on the paper.
Line 1-4: Since Globularia alypum plays an important role in the paper, inclusion of the name in the title will be appropriate and can increase readership.
Line 22: “evaluated for their antibacterial and antibiofilm activities with the microdilution method and antibiofilm effects via the crystal violet assay.” Not clear what this means. What is the difference between antibiofilm activity and antibiofilm effect?
Line 54: Change “associated to” to “associated with”
Line 103: “into three parts as upper, middle” Only two parts are mentioned, the third one is missing.
Line 107: “All segments of the stent parts including inner surface, middle and outer surface” These descriptions of the parts don’t agree with those in line 103. Are these same or different?
Line 131: “antimicrobial categories [21]”. I understand that a reference has been provided, but for the benefit of the readers it will help to mention the categories when all the antibiotics are first mentioned in lines 121-124.
Line 140 Table 1: I have never seen a table with one row of data in it. This can be simply written in the text instead of making a table.
Line 142: “30 min in 100 mL of boiling distilled water”. Was the final volume measured or adjusted to 100 mL? Some water would have been lost during boiling.
Line 143: “lyophilized and stored” Was it lyophilized to dryness? How was stored? As a solid powder? Or as solution?
Line 148: Put “High Performance Liquid Chromatography” in parenthesis “( )”
Line 154: Three times a space has been added before a comma instead of after the comma.
Line 154: Change “Simple” to “Sample”
Line 156: Change “Peaks identificationwas obtained” to “Peak identifications were done”
Line 159: “qualibration” I am not familiar with this word. Hope it is correct.
Line 162: Change “baumanii. The” to “baumanii, the”
Line 163: “mop” I guess, the authors mean “swab”
Line 159: “20 L” I guess, the authors mean “20 ml” (microliter)
Line 166: “The extracts were diluted in sterile distilled water at a concentration of 25 mg/mL” How was the concentration determined? The extract is a solution. Does the 25 mg refer to the weight of the solution or that of the solute in the solution? If it is the solute, how was that measured. This is extremely important for reproducibility and should be mentioned in the methods. The final volume of the extract after boiling for 30 min is also not mentioned in the methods.
Line 168, 341: Use the spelling accepted by the journal: “vancomycine” or “vancomycin”
Line 168: “30 g/mL; 10 L/disc” Here and throughout the manuscript please ensure that the units have been written correctly and that symbol fonts (for example, micro) show properly in the pdf. I will not point out these errors further, but they appear numerous times.
Line 175: Change “found to have” to “reported to be”
Line 180: “BHI” Write the full form when an abbreviation is mentioned for the first time. This is actually done in line 182.
Line 209: “bacillus negative grams and a cocci positive Gram” Never seen it written this way. I would write it as “Gram negative bacilli and a Gram positive cocci”
Line 212: Change “showed several resistance to most” to “showed resistance to most”
Line 216: Change “For P. putida, they were” to “P. putida were”
Line 212-216: The authors have made Table 1 with only one row of data in it while these data on resistance have been presented in text. The best place for this information is in Table 2, where the bacteria are listed.
Line 225: Change “For P. putida; it has been demonstrated that this bacteria caused” to “P. putida has been demonstrated to cause ”
Line 233: Change “sufficient to the bacterial” to “sufficient for bacterial”
Line 234: Change “Biofilm formation on stent colonization was well defined.” to “Biofilm formation on stent has been well-defined.”
Line 240: Change “has been showed” to “has been shown”
Line 242: Change “The urease, enzyme” to “Urease, an enzyme”
Line 243: Change “cleaves urea to ammonia and increased urinary pH results in precipitation of” to “cleaves urea to ammonia, which increases the urinary pH and results in precipitation of”
Line 264: Change “before to stent” to “before stent”
Line 277: Change “limit number of patient” to “limited number of patients”
Line 242: Change “age, or to make a comment” to “age.”
Line 285: Change “compounds classes” to “compound classes”
Line 286: “glycoside or not” Not clear what this means.
Line 292: Change “different to” to “different from”
Line 293: Change “showed” to “have shown”
Line 298: Change “et al” to “et al.”
Line 308: Change “my” to “may”
Line 298: Change “extracts” to “extracts, ”
Line 311: Change “are conducted” to “have been conducted”
Line 312: “Only a few plants have been evaluated for these extracts” Not clear if the authors are discussing G. alypum or any plant. A quick search of water soluble antibiotics from plants shows that there are many papers published.
Line 324, Figure 2: If the idea is to compare the two profiles, the two should be aligned such that the same peaks are the at the same position, one below other.
Line 324: In the top chromatogram, the last two names should be brought closer to or above the arrows.
Line 332: Change “was” to “being”
Line 333: Change “No data was performed” to “No data has been reported”
Line 335: Change “activity” to “activity,”
Line 339: Change “others tested strains” to “other tested strains”
Line 341: The zones of inhibition of vancomycin and infusion appear to be similar. However, zones can always be made to appear similar by adjusting the concentration and volume spotted. Please mention this information either here or in the Methods.
Line 343: Same comment as for line 166. How was the weight of the extract determined? Was it first lyophilized to dryness? Did this have any insoluble cellulosic matter when resuspended in water or buffer?
Line 344: “Nayak et al. [44]” The reference number does not match with that in the list of references. Please check that all your references are correctly numbered.
Line 373: Change “in EPS” to “of EPS”
Line 378: Change “had shown” to “have been shown to have”
Line 378: “antibacterial activities and biofilm” Not clear what this means. Do you mean, “antibacterial and antibiofilm activities”?
Line 384 Table 4: All numbers have superscripts a,b and c but there is no footnote explaining what these represent. Since all numbers in a column have the same superscript, it can be shown on the name of the bacteria rather than on each number.
Line 384: What do the “,” mean? In other tables a decimal has been written as a “.” instead of a “,”
Line 385: Change “;” to “,”
Line 386: Change “pathogen” to “pathogens”
Line 395: Change “related” to “related to”
Line 442: Delete “.” before “Brun”
Line 480: Delete “.” before “Akay”
Quality of English is acceptable. There are some minor errors.
Author Response
Dear Sanja Sokolov,
On behalf of my co-authors, I am pleased to submit the revised version of our manuscript “Analysis of and Pathogens of Urinary Tract Infection Associated with Indwelling Double-J Stent and Their Susceptibility to Globularia alypum.” that was previously submitted to “Pharmaceutics” (referenced as Pharmaceutics-2619262). We would like to thank you and the reviewers for your time and efforts in reviewing our study and for underlining the interest of our findings. We have taken all the reviewer’s remarks and comments into account and significantly revised our manuscript. we provided in the following pages, a point-by-point response to the reviewers’ comments. As recommended, all the modifications are highlighted in yellow in the manuscript.
We hope that this revised version of our manuscript now meets the standards that you and the reviewers recommended, and we respectfully ask you to reconsider it for publication in your journal.
Yours sincerely,
Dr Amira ZAÏRI
Corresponding author
Response to Reviewers’ Comments
Reviewer #1
- The authors demonstrate the antibacterial and antibiofilm activities in Globularia alypum against pathogens of urinary tract infections. One important aspect of this paper is that aqueous extract from the plant has been used while majority of previous investigations have focused on essential oils or methanolic extracts. The paper makes a significant contribution to the field. I have the following additional comments on the paper.
- Line 1-4: Since Globularia alypum plays an important role in the paper, inclusion of the name in the title will be appropriate and can increase readership: We changed the title of the manuscript and included the name of our plant “Globularia alypum” in it. Please refer to page 1, line 3.
- Line 22: “evaluated for their antibacterial and antibiofilm activities with the microdilution method and antibiofilm effects via the crystal violet assay.” Not clear what this means. What is the difference between antibiofilm activity and antibiofilm effect? We modified the sentence to make it more clear. Please refer to page 1, line 22
- Line 54: Change “associated to” to “associated with”: Corrected, please see page 2, line 61.
- Line 103: “into three parts as upper, middle” Only two parts are mentioned, the third one is missing: We added the outer surface to the sentence, please see page 3, line 111.
- Line 107: “All segments of the stent parts including inner surface, middle and outer surface” These descriptions of the parts don’t agree with those in line 103. Are these same or different? The segments of the stent parts used in this study are the same. We corrected them as it was recommended. please see page 3, line 111.
- Line 131: “antimicrobial categories [21]”. I understand that a reference has been provided, but for the benefit of the readers it will help to mention the categories when all the antibiotics are first mentioned in lines 121-124: Categories of antibiotics are added to the current paragraph. Please see page 3 line 129.
- Line 140 Table 1: I have never seen a table with one row of data in it. This can be simply written in the text instead of making a table: We removed the table from our manuscript and all the information regarding geographic localization were written in the text. Please see page 3, line 147.
- Line 142: “30 min in 100 mL of boiling distilled water”. Was the final volume measured or adjusted to 100 mL? Some water would have been lost during boiling: 100 ml was the final volume. In fact, as soon as the water boils, we add our extract, so no risk to lose water during this experiment. The cited protocol belongs to the infusion method as it was described on bibliography..
- Line 143: “lyophilized and stored” Was it lyophilized to dryness? How was stored? As a solid powder? Or as solution? The extract was stored as a solid powder. This information was added to the article, please see page 4, lines 155-156..
- Line 148: Put “High Performance Liquid Chromatography” in parenthesis “( )”: We replaced the name of the chromatography by its abbreviation since it was totally cited in the title of the section. Please see page 4, lines 160.
- Line 154: Three times a space has been added before a comma instead of after the comma. All spaces were removed. Please see page 4, line 166.
- Line 154: Change “Simple” to “Sample”. Page 4, line 166.
- Line 156: Change “Peaks identification was obtained” to “Peak identifications were done”. Please see page 4, line 168.
- Line 159: “qualibration” I am not familiar with this word. Hope it is correct. We changed “qualibration” to “Calibration”. Please see page 4, line 171.
- Line 162: Change “baumanii. The” to “baumanii, the”. Please see page 4, line 170.
- Line 163: “mop” I guess, the authors mean “swab”. We changed “mop” to “swab” page 4, line 175.
- Line 159: “20 L” I guess, the authors mean “20 ml” (microliter). Due to different version of “word software”, units were changed during the submission process. We corrected all the units in the whole manuscript. Please see for example, page 4, line 180.
- Line 166: “The extracts were diluted in sterile distilled water at a concentration of 25 mg/mL” How was the concentration determined? The extract is a solution. Does the 25 mg refer to the weight of the solution or that of the solute in the solution? If it is the solute, how was that measured. This is extremely important for reproducibility and should be mentioned in the methods. The final volume of the extract after boiling for 30 min is also not mentioned in the methods. 25 mg/ml refer to the weight of our extract solubilized in 1 ml. The 25 mg of our extract were obtained after lyophilization procedure. It was mentioned in our article. However, we added “lyophilized extract” to mention that this concentration was prepared after aqueous extraction and lyophilization. Please see page 4, line 179.
- Line 168, 341: Use the spelling accepted by the journal: “vancomycine” or “vancomycin”. We are so sorry for this mistake. In fact, we used Imipenem and not vancomycin. The name of the antibiotic was then corrected in the whole manuscript. Please see page 4, line 180.
- Line 168: “30 g/mL; 10 L/disc” Here and throughout the manuscript please ensure that the units have been written correctly and that symbol fonts (for example, micro) show properly in the pdf. I will not point out these errors further, but they appear numerous times. We corrected units throughout the manuscript.
- Line 175: Change “found to have” to “reported to be”. Please see page 4 line 190.
- Line 180: “BHI” Write the full form when an abbreviation is mentioned for the first time. This is actually done in line 182. We wrote the full form of BHI. Please see page 5, line 197.
- Line 209: “bacillus negative grams and a cocci positive Gram” Never seen it written this way. I would write it as “Gram negative bacilli and a Gram-positive cocci”. We are sorry for this inconvenience, and we correct the bacteria names as it was recommended. Please see page 5, line 226.
- Line 212: Change “showed several resistance to most” to “showed resistance to most”. Please see page 5, line 229.
- Line 216: Change “For putida, they were” to “P. putida were”. Corrected. Please see page 5, line 229.
- Line 212-216: The authors have made Table 1 with only one row of data in it while these data on resistance have been presented in text. The best place for this information is in Table 2, where the bacteria are listed. We included the data on resistance in the table. Please see page 6, line 247.
- Line 225: Change “For putida; it has been demonstrated that this bacteria caused” to “P. putidahas been demonstrated to cause”. Corrected, please see page 5, line 242.
- Line 233: Change “sufficient to the bacterial” to “sufficient for bacterial” and Line 234: Change “Biofilm formation on stent colonization was well defined.” to “Biofilm formation on stent has been well-defined.” These two sentences were corrected, but they were included in the introduction section as it was recommended by Reviewers 2. Please see page 2, lines 54-60.
- Line 240: Change “has been showed” to “has been shown”. Please see page 6, line 261.
- Line 242: Change “The urease, enzyme” to “Urease, an enzyme”. Please see page 6, line 263.
- Line 243: Change “cleaves urea to ammonia and increased urinary pH results in precipitation of” to “cleaves urea to ammonia, which increases the urinary pH and results in precipitation of”. Please see page 6, line 265.
- Line 264: Change “before to stent” to “before stent”. Please see page 6, line 286.
- Line 277: Change “limit number of patient” to “limited number of patients”. Please see page 7, line 299.
- Line 242 (Line 278): Change “age, or to make a comment” to “age.” Please see page 7, line 300.
- Line 285: Change “compounds classes” to “compound classes” Please see page 7, line 310.
- Line 286: “glycoside or not” Not clear what this means. Please see page 7, line 311.
- Line 292: Change “different to” to “different from” Please see page 7, line 317.
- Line 293: Change “showed” to “have shown” Please see page 7, line 318.
- Line 298: Change “et al” to “et al.” Please see page 7, line 324.
- Line 308: Change “my” to “may”. Please see page 7, line 334.
- Line 298: Change “extracts” to “extracts, ” Please see page 7, line 335.
- Line 311: Change “are conducted” to “have been conducted”. Please see page 7, line 337.
- Line 312: “Only a few plants have been evaluated for these extracts” Not clear if the authors are discussing G. alypum or any plant. A quick search of water-soluble antibiotics from plants shows that there are many papers published. We corrected the sentence to mention that the majority of plants including G. alypum are evaluated for their biological activities as organic extracts. Infusion and decoction, which are considered as aqueous extracts are less investigated. Please see page 8, line 339.
- Line 324, Figure 2: If the idea is to compare the two profiles, the two should be aligned such that the same peaks are the at the same position, one below other. Unfortunately, we failed to align figure such that the same peaks will be in the same position. To overcome this point, we added numbers on the peaks to mention that they are the same in the two figures. Please see page 8, lines 344-346..
- Line 324: In the top chromatogram, the last two names should be brought closer to or above the arrows. We adjust the position of arrows, so they correctly show the corresponding compound. Please see page 8, line 344.
- Line 332: Change “was” to “being”. Please see page 8, line 353.
- Line 333: Change “No data was performed” to “No data has been reported”. Please see page 8, line 356.
- Line 335: Change “activity” to “activity,” Please see page 8, line 359.
- Line 339: Change “others tested strains” to “other tested strains”. Please see page 9, line 363.
- Line 341: The zones of inhibition of vancomycin and infusion appear to be similar. However, zones can always be made to appear similar by adjusting the concentration and volume spotted. Please mention this information either here or in the Methods. The information cited above was added. Please see page 8, line
- Line 343: Same comment as for line 166. How was the weight of the extract determined? Was it first lyophilized to dryness? Did this have any insoluble cellulosic matter when resuspended in water or buffer? As it was mentioned in the comment (Line 166), information regarding the concentration of our extract were added. Please see page 4, line 179. It’s important to notice that our extract was very soluble in distilled water and no insoluble cellulosic issue was observed.
- Line 344: “Nayak et al. [44]” The reference number does not match with that in the list of references. Please check that all your references are correctly numbered. The correct number of the reference is added. Please see page 9, line 367.
- Line 373: Change “in EPS” to “of EPS” Please see page 9, line 396.
- Line 378: Change “had shown” to “have been shown to have” Please see page 10, line 401.
- Line 378: “antibacterial activities and biofilm” Not clear what this means. Do you mean, “antibacterial and antibiofilm activities”? We mean “antibacterial and antibiofilm activities”. We correct the sentence, please see page 10, line 402.
- Line 384 Table 4: All numbers have superscripts a,b and c but there is no footnote explaining what these represent. Since all numbers in a column have the same superscript, it can be shown on the name of the bacteria rather than on each number. A footnote explaining the superscripts a, b and c is added and they are shown on the name of the bacteria. Please see page 10, line 407.
- Line 384: What do the “,” mean? In other tables a decimal has been written as a “.” instead of a “,”. We corrected decimals in the table, please see page 10, line 410.
- Line 385: Change “;” to “,”. Corrected, please see page 10, line 411.
- Line 386: Change “pathogen” to “pathogens”. Corrected, please see page 10, line 412.
- Line 395: Change “related” to “related to” Corrected, please see page 10, line 422.
- Line 442: Delete “.” before “Brun” and Line 480: Delete “.” before “Akay”. “.” Were deleted in both references. Page 11, line 468 and line 506.
Thanks a lot for your comments and efforts.
Reviewer 2 Report
The work presents a microbiological studies of DJ stents and an analysis of the impact of compounds of plant origin on pathogens colonizing these biomaterials. Due to the growing antibiotic resistance of bacteria and complications related to the use of biomaterials, the topic is important and fits into a very current trend of research.The authors analyzed almost 60 stents and showed the presence of only three strains of bacteria. The analyzed extracts of Globularia alypum L. showed antimicrobial and antibiofilm activity against the isolated bacteria, however, to a small extent and in quite high concentrations. However, the work has methodological shortcomings and lacks certain information. Below are my comments and remarks
- title of the manuscript - I think it would be more understandable in the following form „ Analysis of of Pathogens of Urinary Tract Infection Associated with Indwelling Double-J Stent and Their Susceptibility to Natural Compounds” the word „characteristics” is redundant here
- abstract - the authors state in the text „They can, develop bacterial encrustation …” Encrustation is the deposition of various components on the surface, including minerals, as the authors stated later in the text, but not of living organisms such as bacteria. Colonization, adhesion etc. would be a more accurate formulation
- introduction – has been written „…biofilm bacteria..” it should be - bacteria living in a biofilm or bacteria forming a biofilm
- introduction (line 53) – „..Biofilm bacteria differ physiologically and phenotypically from planktonic or suspended cells”. Please specify what these differences are, are they important from the point of view of the research undertaken and their purpose?
- Methods - stated in the text that (line 102) „The removed DJ stents were …stored in 50 mL sterile centrifuge tubes… at 4 °C for a maximum of 5 days” - why so long? Couldn't this affect the viability of the bacteria?
- Methods – lines 166 -172 „..and 20 L of each extract were deposited onto each disc..”.and „…As a standard antibiotic, vancomycine (30 g/mL; 10 L/disc) was employed” and „The MBC was assessed by transferring 10 L from the well that showed „- Is this unit correct and is it liters and g?
- Methods - How was the MIC assessed, was it assessed through observation or spectrophotometric measurement, were there controls on bacterial growth without extract?
- how were extract solutions prepared for testing? How the specific concentration was obtained in mg/ml. Were the extracts at concentrations of 25 mg/ml still highly soluble? Were extracts obtained once used in the research, or were they obtained several times and standardized?
- line 208 „..bacillus negative grams..” – sholud be- Gram-negative bacilli
- lines 231-279 - the information contained in this fragment should be included in the introduction of the work, not in the results.
- Table 4 lacks information that it represents the percentage of inhibition
- With such a small number of isolated bacteria, the authors cannot conclude that A. baumannii; E. faecalis and P. putidis were the most frequently isolated pathogens. (line 385)
- There is no critical approach to microbiological research in the text of the work. Have studies in other centers resulted in a similar isolation frequency and profile of isolated bacteria?
- Research on extracts should be supplemented with a broader panel of microorganisms, then conclusions regarding their impact on microorganisms would be more justified.
Author Response
Reviewer #2
The work presents a microbiological studies of DJ stents and an analysis of the impact of compounds of plant origin on pathogens colonizing these biomaterials. Due to the growing antibiotic resistance of bacteria and complications related to the use of biomaterials, the topic is important and fits into a very current trend of research.The authors analyzed almost 60 stents and showed the presence of only three strains of bacteria. The analyzed extracts of Globularia alypum L. showed antimicrobial and antibiofilm activity against the isolated bacteria, however, to a small extent and in quite high concentrations. However, the work has methodological shortcomings and lacks certain information. Below are my comments and remarks
- Title of the manuscript - I think it would be more understandable in the following form „Analysis of Pathogens of Urinary Tract Infection Associated with Indwelling Double-J Stent and Their Susceptibility to Natural Compounds” the word „characteristics” is redundant here. We changed the title of manuscript based on the reviewer’s proposition. Please see page 1, line 2-3.
- Abstract - the authors state in the text„They can, develop bacterial encrustation …” Encrustation is the deposition of various components on the surface, including minerals, as the authors stated later in the text, but not of living organisms such as bacteria. Colonization, adhesion etc. would be a more accurate formulation. We corrected the abstract and we would like to thank a reviewers for this clarification. Please see page 1, line 18.
- Introduction– has been written „…biofilm bacteria..” it should be - bacteria living in a biofilm or bacteria forming a biofilm. We changed the sentence as recommended. Please see page 2, line 53.
Introduction (line 53) – „..Biofilm bacteria differ physiologically and phenotypically from planktonic or suspended cells”. Please specify what these differences are, are they important from the point of view of the research undertaken and their purpose?. We added a paragraph explaining the difference between, planktonic bacteria and biofilm. This is very important for our research since our extract showed an important biofilm effect. In fact, several studies showed biofilm are very resistant to antibiotics which are effective against planktonic bacteria. Please see page 2, line 53.
Methods - stated in the text that (line 102) „The removed DJ stents were …stored in 50 mL sterile centrifuge tubes… at 4 °C for a maximum of 5 days” - why so long? Couldn't this affect the viability of the bacteria?. Basically, Dj stents were sent directly to laboratory of bacteriology for analysis. However, sometimes and due to technical problems, devices are not assessed directly and kept for 5 days. The choice of maximum 5 days was conducted based on bibliography.
Methods – lines 166 -172 and 20 L of each extract were deposited onto each disc..”.and „…As a standard antibiotic, vancomycine (30 g/mL; 10 L/disc) was employed” and „The MBC was assessed by transferring 10 L from the well that showed „- Is this unit correct and is it liters and g?. We are sorry for this inconvenience, in fact, during submission procedure, some words were changing in word files. We corrected all units throughout the manuscript. They were generally, µl/mL.
- Methods - How was the MIC assessed, was it assessed through observation or spectrophotometric measurement, were there controls on bacterial growth without extract?. Additional data were included in methods section (antibacterial activity) to better explain the assessment of MICs. Please see page 4, lines 185 and 192.
- how were extract solutions prepared for testing? How was the specific concentration obtained in mg/ml. Were the extracts at concentrations of 25 mg/ml still highly soluble? Were extracts obtained once used in the research, or were they obtained several times and standardized? After extraction and lyophilization, we obtain a solid powder. To prepare a concentration 25mg/ml, we measure the weight (25 mg) and dissolve it in 1 ml. The obtained extract was very soluble since it was extracted by infusion method. This protocol was repeated every time we have experimental tests are needed.
Line 208 „..bacillus negative grams..” – sholud be- Gram-negative bacilli. Corrected. Please see page 5, line 226.
- lines 231-279 - the information contained in this fragment should be included in the introduction of the work, not in the results. We included the indicated part in the introduction as it was recommended. Please see page 2, line 54-line 60.
Table 4 lacks information that it represents the percentage of inhibition. Percentage symbol (%) was included in the table to mention that inhibition of biofilm is showed by percentage unit. Please page 10, line 409.
With such a small number of isolated bacteria, the authors cannot conclude that A. baumannii; E. faecalis and P. putidis were the most frequently isolated pathogens. (line 385). We agree with the reviewer’s comment, and we removed “most frequently” from the text. Please see page 10, line 411.
There is no critical approach to microbiological research in the text of the work. Have studies in other centers resulted in a similar isolation frequency and profile of isolated bacteria? Unfortunately, due to the lack of similar studies in other centers, we can not confirm that we have the similar isolation frequency and profile of isolated bacteria. This point was mentioned in the manuscript.
- Research on extracts should be supplemented with a broader panel of microorganisms, then conclusions regarding their impact on microorganisms would be more justified. We would like to thank the reviewer for this comment. In fact, including additional data regarding our extract will be beneficial, but due the limited number of words recommended by the journal we cannot develop more in discussion section. In addition, only few studies are conducted regarding the antibacterial activity of aqueous extracts of G. alpym which make the discussion part limited.
Thanks a lot for your comments and efforts.
Reviewer 3 Report
The manuscript describes the analysis of characteristics and pathogens of urinary tract infection associated with indwelling double-J stent and their susceptibility to natural compounds. The topic is relevant to the aim and scope of the Pharmaceutics. The manuscript is well written and easy to follow. Some clarifications in the texts are needed. Overall, this manuscript meets the standard for acceptance after addressing the below comments:
1) Please describe in the manuscript the reason why only the effect of vancomycine was shown. How about the effect for Gram-negative strains?
2) Please describe in the manuscript the reason why the Reverse phase HPLC instead of normal HPLC.
3) Among the Gram-positive strains, why have E. faecalis, P. putida, and A. baumanii been chosen rather than other strains for this research? Are they the most popular for the urinary tract infection associate the stent? If then, please add this fact in the manuscript with references.
4) In Table 5, MIC of A. baumanii has been found to be lower than the other two. Why should MIC of A. baumanii be the lowest?
5) There are two Table 4s. I believe that the latter would be Table 6. Comparing Table 5 and 6, the MIC of A. baumanii in Table 5 is lower while its percentage in Table 6 is higher.
Author Response
Reviewer #3
The manuscript describes the analysis of characteristics and pathogens of urinary tract infection associated with indwelling double-J stent and their susceptibility to natural compounds. The topic is relevant to the aim and scope of the Pharmaceutics. The manuscript is well written and easy to follow. Some clarifications in the texts are needed. Overall, this manuscript meets the standard for acceptance after addressing the below comments:
- Please describe in the manuscript the reason why only the effect of vancomycine was shown. How about the effect for Gram-negative strains? We would like to thank the reviewer for this comment since we forgot to correct the used antibiotic before submission. In fact, we used the imipenem and not vancomycin. This error is corrected throughout the manuscript.
- Please describe in the manuscript the reason why the Reverse phase HPLC instead of normal HPLC. A paragraph and reference were added to explain why we used RP-HPLC instead of HPLC. Please see page 7, lines 303-307.
- Among the Gram-positive strains, why have faecalis,P. putida, and A. baumanii been chosen rather than other strains for this research? Are they the most popular for the urinary tract infection associate the stent? If then, please add this fact in the manuscript with references. E. faecalis, P. putida, and A. baumanii are not chosen, they were isolated and identified from the DJ stents that were removed from 59 patients. Thus, we were obliged to evaluate the antimicrobial activities of our extracts againt these strains. All these information is well mentioned in all the manuscript’s sections.
- In Table 5, MIC of baumaniihas been found to be lower than the other two. Why should MIC of A. baumanii be the lowest? Unfortunately, we are unable to explain why our extract was more effective against A. baumanii than the other strains. Further studies are warranted in order to know the mechanism by which aqueous extract act. Among these studies , we cite for example electronic microscope and mass spectrometry, these methods are very costly and we do not have them in our laboratory.
- There are two Table 4s. I believe that the latter would be Table 6. Comparing Table 5 and 6, the MIC of baumaniiin Table 5 is lower while its percentage in Table 6 is higher. Number of tables was corrected in the whole manuscript.
Thanks a lot for your comments and efforts.
Round 2
Reviewer 2 Report
I would like to thank the authors for their replies and changes made to the manuscript. However, there is still no information on the differences between planktonic and biofilm cells. This is important information from the point of view of the research topic. Bacteria in a biofilm are physiologically different from planktonic bacteria, which, together with the structure of the biofilm, makes them more resistant to antimicrobial agents, why? This information should be provided because the study also compares the antimicrobial effect of the extract on both bacterial forms. The revised fragment only contains information on biofilm formation.
Furthermore, please correct it
-the word "and" is unnecessary in the title of the manuscript
- the title of table 2 should be bacterial strains and not colonies
- line 282 - it should be staphylococci or Staphylococcus sp.
Author Response
On behalf of the authors, firstly, we would like to thank you for your efforts and for all your modifications to make our work suitable for publication.
1- As you recommended, we added a paragraph explaining the differences between planktonic and biofilm cells. Please see page 1, line 54.
2- the word "and" is unnecessary in the title of the manuscript: We removed the word "and" in the title. Please see page 1, line 2.
3- The title of table 2 should be bacterial strains and not colonies: The title of table 2 was changed as recommended. Please see page 6, line 254.
4- line 282 - it should be staphylococci or Staphylococcus sp. We changed staphylococcus to Staphylococcus sp. Please see page 7, line 278.
Reviewer 3 Report
The issues have been addressed.
Author Response
On behalf of the authors, we would like to thank you for your efforts and for all your modifications to make our work suitable for publication.
Warm regards